

# Mid-Holocene reinforcement of North Atlantic atmospheric circulation variability from a western Baltic lake sediment record

Markus Czymzik[1], Rik Tjallingii[2], Birgit Plessen[2], Peter Feldens[1], Martin Theuerkauf[3], Matthias Moros[1], Markus J. Schwab[2], Carla K.M. Nantke[1], Silvia Pinkerneil[2], Achim Brauer[2,4], Helge W. Arz[1]

[1]Leibniz Institute for Baltic Sea Research Warnemünde (IOW) - Marine Geology, Rostock, Germany

[2]GFZ German Research Centre for Geosciences - Climate Dynamics and Landscape Evolution, Potsdam, Germany

[3]University of Greifswald - Institute of Botany and Landscape Ecology, Greifswald, Germany

[4]University of Potsdam - Institute of Geosciences, Potsdam, Germany

*Correspondence to:* Markus Czymzik (markus.czymzik@io-warnemuende.de)

**Abstract.** Knowledge about timing, amplitude and spatial gradients of Holocene environmental variability in the Circum-Baltic region is key to understand its responses to ongoing climate change. Based on a multi-dating and proxy approach, we reconstruct changes in productivity from TOC contents in sediments of Lake Kälksjön (KKJ) in west-central Sweden spanning the last 9612 (+255/-144) years. An exception is the period from AD 1878 until today, in which sedimentation was

dominated by anthropogenic lake level lowering and land use. In-lake productivity was higher during periods of warmer winters with shortened ice cover and prolonged growing seasons. A multi-millennial increase in productivity throughout the last ~9600 years is associated with progressively warmer winters in north-western Europe, likely triggered by the coinciding increase in Northern Hemisphere winter insolation. Decadal to centennial periods of higher productivity in KKJ correspond to warmer winters during a more positive North Atlantic Oscillation (NAO) polarity, as reconstructed for the last 5200 years.

In consequence, we assume our decadal to centennial productivity record from KKJ sediments for the complete ~9600 years to provide a qualitative record of NAO polarity. A shift towards higher productivity variability at ~5450 cal. a BP is hypothesized to reflect a reinforcement of NAO-like atmospheric circulation variability, possibly driven by more vigorous changes in North Atlantic deep water formation.


## 1. Introduction

The Circum-Baltic ecosystems are sensitive to both natural climate forcing and human impact (Andrén et al., 2000; BACC II, 2015; Warden et al., 2017). Existing studies from these regions point to considerable paleoenvironmental variability and spatial gradients active on inter-annual to millennial time-scales, triggered by a complex interplay of internal and external

forcing mechanisms (BACC II, 2015; Moros et al., 2020; Zillén et al., 2008). Particularly the north-western 'Scandinavian' region of the Baltic realm experienced a unique Holocene development. Postglacial melting of the Fennoscandian ice sheet led to isostatic rebound of up to 300 m, eustatic sea level changes and shoreline displacements (Berglund, 2004; Hansson et al., 2018). In addition, the sheltering effect of the Scandinavian Mountains North of 62°N results in varying expressions of the prevailing North Atlantic climate (Barnekow, 2000; Blenckner et al., 2004; Hammarlund et al., 2003; Snowball et al.,

1999). Existing paleoenvironmental information provide little detail about the possible ranges and gradients of atmospheric circulation changes and a better delimitation of such changes can provide a more comprehensive understanding of past natural climate variability in the western Baltic region and, ultimately, Circum-Baltic climate change. Lake sediment records



can provide this type of information. The composition and geochemical features of their sediment columns is modulated by past climate and anthropogenic changes in their catchment (Labuhn et al., 2018; Zolitschka et al., 2015).

Lake Kälksjön (KKJ) in west-central Sweden is well-suited for the reconstruction of changes in synoptic atmospheric circulation for most of the Holocene. Previous studies have proven the KKJ sediment column a continuous archive of secular changes in Earth's magnetic field and winter precipitation connected with the 8.2 ka cold event (Randsalu-Wendrup et al., 2012; Snowball et al., 2010; Stanton et al., 2010, 2011). Its position in central Sweden at 60°N is sensitive to changes in the North Atlantic climate and not substantially affected by the orographic shielding of the Scandinavian mountains (Hurrell,

1995; Uvo, 2003). In this study, we apply a multi-proxy and dating approach to reconstruct changes in lake productivity and deduce the driving climate and human factors throughout the last ~9600 years.

## 2. Study site

KKJ is located in west-central Sweden, at an altitude of 97 m a.s.l. (60°09'N/13°02'E) (Fig. 1) (Stanton et al., 2010; Zillén et al., 2003). The modern basin was isolated from ancient Lake Vänern through isostatic rebound following the last deglaciation. KKJ has a surface area of 0.42 km$^2$, maximum water depth of 13.6 m and catchment size of 4 km$^2$ (Fig. 1) (Stanton et al., 2011; Zillén et al., 2003). Four creeks discharge into the lake from the hilly and forested north-eastern shores. It's only outflow in the West was artificially incised in AD 1878 to lower the lake level and enlarge the grassland for

stockbreeding in the southern and western catchment (Fig. 2). In October 1993 and 1999, 64.2 and 67.3 tons of chalk flour were distributed in the lake to raise the water pH level, respectively (Stanton et al., 2010).

Today's climate at KKJ is controlled by its location at the transition between the temperate and subpolar zones. Mean monthly temperatures vary between 16°C in July and -6°C in January (Fig. 3). Lake ice cover in the KKJ area usually lasts from December to April (Blenckner et al., 2004). Annual mean precipitation is 720 mm with a maximum from June to

October (Fig. 3).

## 3. Material and methods

### 3.1. Hydroacoustic survey

Hydroacoustic bathymetry surveys were performed using a portable Norbit iWBMSe device with a central frequency of 400 kHz and 80 kHz bandwidth. Data were recorded with the Norbit GUI in s7k format and processed using mbsystem (Caress and Chayes, 1996), correcting for sound velocity, manually removing outliers and creating grids of 0.5 m resolution. Seismic data were recorded using a parametric Innomar 96 sediment echo sounder, with the low frequency set to 10 kHz and a vertical resolution of < 6 cm. Data were processed and plotted with Seismic Unix applying a bandpass filter, binning to 1 m

intervals and a time-varied gain (Stockwell, 1999). Two-way travel times were converted to meters applying a sound velocity of 1500 m s$^{-1}$. Positional data for the bathymetric and seismic surveys were provided by a POSMV Seastar system working with the EGNOS correction and a spatial accuracy of ~45 cm.

### 3.2. Composite profile KKJ19

Parallel and overlapping sediment cores KKJ19-A and KKJ19-B were retrieved in August 2019 from the deepest part of KKJ using an UWITEC piston corer. The cores consist of 2 m long segments, with a diameter of 9 cm. Two surface sediment cores were retrieved from the same position with an UWITEC short core system to reduce process-based disturbances of the water soaked sediment surface. Composite profile KKJ19 was constructed through core-to-core correlation of macroscopic lithological layers and has a total length of ~8 m.




### 3.3. Microfacies analyses

Microfacies analyses of varying sediment properties were performed on a series of overlapping petrographic thin-sections (10×2 cm) from composite profile KKJ19. Microscopic investigations were performed using a ZEISS Axiolab polarization microscope at 25 x to 200 x magnification, under varying light and optical conditions. Particle size measurements were

performed at 200 x magnification. Thin-section preparation from unconsolidated sediments includes shock-freezing with liquid nitrogen, freeze-drying and epoxy resin impregnation under vacuum (Brauer and Casanova, 2001; Czymzik et al., 2018).

### 3.4. Geochemistry

Total organic carbon (TOC) contents, $\delta^{13}C$ values of organic matter ($\delta^{13}C_{org}$) and C/N-ratios were measured at 1 cm resolution from freeze dried and homogenized sediment samples using an EA Isolink elemental analyzer coupled to a DELTA V advantage isotope ratio mass spectrometer (Thermo Fisher Scientific). Before the measurements, the samples were weighted into Ag capsules, in-situ decalcified, first with 3% and second with 20% HCl and dried for 3 hours at 75°C. The calibration was performed based on an elemental (Urea) and certified isotope standard (IAEA-CH-7: $\delta^{13}C$ -31.8‰) and

checked with an internal soil reference sample (Boden3, HEKATECH). Replicate analyses of the standards resulted in a reproducibility of 0.2 wt.% for TOC and 0.2‰ for $\delta^{13}C_{org}$.

Element intensity profiles were acquired every 200 µm from the cleaned split-core surface using an ITRAX X-ray fluorescence core scanner (Croudace et al., 2019). Measurements were conducted with a Cr X-ray source operated at 30 kV and 60 mA that irradiated a surface of 0.2 × 8 mm for 3 seconds at each sample position. Element intensities are acquired in

counts per second (cps) for the elements Al, Si, S, K, Ca, Ti, V, Mn and Fe. Additionally, two-fold replicate measurements were acquired for each core section in 2 to 3 intervals of at least 2 cm length. Replicate positions were selected to cover the main sedimentological compositions.

Element correlations are visualized in a principal component (PC) biplot after centred-log ratio transformation of the XRF intensity data. Centred log-ratio transformation of element intensities lifts the compositional constraints and permits rigorous

statistical analyses (Aitchison, 1982, 1986). For this advantage, centred log-ratios of element intensities are used for principal component analyses and calculating groups of similar compositions by ward's hierarchical clustering (Martin-Puertas et al., 2017). Replicate measurements were used for estimating confidence limits and scaling the statistical analyses to reduce noise. All statistical analyses were performed using the Xelerate software package (Weltje et al., 2015).

### 3.5. Pollen analysis


Pollen sample preparation followed the procedure described by Fægri and Iversen (1989) and includes treatment with 25% HCl, 10% KOH, acetolysis at 100° C and 40% cold HF. Samples were finally washed with ethanol and transferred into silicon oil. Pollen samples were analysed at 400 x magnification with a Zeiss-Axiolab microscope. To estimate pollen concentrations and accumulation, three tablets with exotic marker grains (Lycopodium clavatum spores, Batch Nr.

050220211) were added to each sample before preparation. Pollen analyses in KKJ sediments were carried out for the interval from 0 to 63 cm composite depth to cover the main period of human activity and 260 to 410 cm composite depth including the transition from sediment deposition unit (SDU) 3 to 4 (see the results).

### 3.6. Chronology

An age-model for composite profile KKJ19 was calculated using the Oxcal software working with a P-sequence model and the IntCal20 calibration curve (Bronk Ramsey, 2008; Reimer et al., 2020). The calculations were carried out based on $^{14}C$ ages of plant macrofossils and lithological marker layers of known age. Additional measurements of $^{137}Cs$ and $^{241}Am$



contents were performed with a Canberra Ge-detector BE3830-7500SL-RDC-6-ULBB gamma spectrometer (Moros et al., 2017).


### 3.7. Time-series analysis

Spectral properties of the KKJ TOC record and a multi-millennial NAO reconstruction (Olsen et al., 2012) were computed using fast Fourier transform based on least-square fitting of sinusoids to the time-series (Ghil et al., 2002). Spectral significance levels were calculated using 10000 iterations of an autoregressive model fitted to the original data. Prior to the

analysis, the time-series were resampled to a 20-year resolution and detrended.

## 4. Results

### 4.1. Bathymetry

KKJ is separated into a 6.1 m deep southern and 13.6 m deep central basin, divided by a sill with ~4 m water depth (Fig. 1). The deepest part of the central basin and coring location of composite profile KKJ19 is flat with slopes < 1°, has a maximum E-W extension of 300 m and N-W extension of 330 m (Fig. 1). A single elevation with 9.8 m water depth is located in the basin's northern part. The flat part of the deep basin reaches within 50 m to the western shore, resulting in slope angles of >20°. There, morphological remnants of a minor landslide extend 60 m into the lake (Fig. 1).


### 4.2. Sediment deposition units

Composite profile KKJ19 can be subdivided into six sediment deposition units (SDU). Boundaries between the SDU are mainly characterized by shifts in organic and detrital matter contents (Figs. 4 and 5). In the deep central basin including the coring location, the seismic data is masked by the presence of free gas (Fig. 1). In consequence, we connect the seismic

records from the basins onset and margin, without masking gas, with the SDU in sediment core KKJ19 (Fig. 1).

*SDU 1 (772 to 600 cm composite depth):* SDU 1 is composed of homogenous clay- to silt-sized detrital grains (Fig. 5), including quartz, feldspar and mica. The predominance of the detrital material is reflected by high Ti values, low TOC contents of ~1% and $\delta^{13}C_{org}$ values up to -25‰ (Fig. 4).


*SDU 2 (600 to 583 cm composite depth):* SDU 2 is characterized by a progressively increasing deposition of organic material, from sporadic layers to a massive organic-rich unit and reduced abundances of detrital grains (Fig. 5). Accordingly, SDU 2 depicts an increase in TOC contents from 2 to 4% and a drop in Ti values (Fig. 4). While the change from SDU 2 to the overlying SDU 3 appears sharp at 583 cm composite depth in the microscopic observations, the drop in Ti points to a

slightly thicker SDU 2 reaching up to 570 cm composite depth (Fig. 4). SDU 1 and 2 likely correspond to the upper part of a seismic unit situated on top of the acoustic base with a chaotic internal texture and few internal laminations. The top of this unit is marked by a high-amplitude seismic unconformity (Fig. 1).

*SDU 3 (583 to 287 cm composite depth):* KKJ sediments in SDU 3 are predominantly composed of amorphous and few

particulate organic material, with some incorporated benthic diatoms and crysophyte cysts (Fig. 5). Within the course of SDU 3, TOC contents increase from 4% at 583 cm to 12% at 287 cm composite depth, superimposed by cm-scale fluctuations (Fig. 4). Low $\delta^{13}C_{org}$ values increase from below -32‰ to up to -30‰, broadly in-phase with the TOC time-series (Fig. 4). Excluding minor peaks, C/N-ratios vary between 14 and 16 (Fig. 4).





*SDU 4 (287 to 37 cm composite depth):* Sedimentological and geochemical features of SDU 4 are comparable to those of SDU 3. Main distinction from SDU 3 are enhanced cm-scale proxy fluctuations (Fig. 4). TOC contents show a further increase with maxima of ~20% in the upper part of SDU 4 (Fig. 4). Even though less distinct, $\delta^{13}C_{org}$ and S/Ti values resemble the trend present in the TOC record (Fig. 4). C/N-ratios within SDU 4 range from about 11 to 14 (Fig. 4). Non-arboreal pollen (NAP) upland and wild grass group pollen show a minor increase at the transition from SDU 3 to 4, but then

return to low values (Fig. 6). A more distinct increase in these settlement indicators occurs at 55 cm composite depth (Fig. 6). SDU 3 and 4 are not readily differentiated in the seismic data, and correspond to a unit with parallel internal laminations that is widespread in the northern basin. Its maximum observed thickness is 5 m, but it is expected that the unit thickness increases in the deepest part of the basin and the core location, which could not be imaged due to the presence of free gas. The unit onlaps to morphological elevations within the northern basin, where its thickness decreases to less than

1 m (Fig. 1).

*SDU 5 (37 to 12 cm composite depth):* Detrital matter contents in SDU 5 increase from 37 cm, reach a maximum around 20 cm and decrease back to background values until 12 cm composite depth (Fig. 4). Littoral diatoms and organic macro-remains are incorporated in the detrital matrix (Fig. 5). Maximum detrital grain size is 50 μm. Variations of detrital matter

abundance as indicated by the Ti record vary anti-phased with the TOC contents that are decreasing from 18% to 4%, before rising back to 16% (Fig. 4). At the onset of SDU 5, wild grass group and NAP upland pollen percentages, as well as the abundance of micro-charcoal particles >25 μm show a distinct increase (Fig. 6).

*SDU 6 (12 to 0 cm composite depth):* KKJ sediments in SDU 6, in general, resemble the organic-rich deposits in SDU 4,

with high TOC contents of about 15% and low Ti values (Fig. 4). One exception is the peak in C/N-ratios up to 42, corresponding to the liming layers (Fig. 4). Wild grass group and NAP upland pollen percentages, as well as the abundance of micro-charcoal particles remain high (Fig. 6). SDU 5 and 6 correspond to the uppermost homogeneous section of the seismic sequence. It is separated from the underlying sediments by a high-amplitude seismic reflection (Fig. 1). The unit is best observed at the basin's margins, but can be traced into the central basin (Fig. 1).


Our stratigraphic sub-division is in general agreement with that of the KKJ sediment core investigated by Stanton et al. (2010). Minor differences in sedimentation rates are likely attributable to slightly different coring locations (~100 m distance) and dissimilar coring techniques (Stanton et al., 2010). By contrast to the results of Stanton et al. (2010), our investigations did not reveal annually laminated (varved) sediments for most of composite profile KKJ19. Possible reasons

might be slight lateral changes in varve preservation within the central basin of KKJ and/or sediment micro-disturbances caused during thin-section preparation. In addition, their varve counts from modified photographs might have revealed sedimentological features that are not visible in our microscopic observations.

**4.3. Geochemical clusters**

Element correlations and clustering results are visualized in a principal component (PC) biplot showing that PC1 and PC2 explain 86.3% and 7.7% of the variance, respectively (Fig. 7). Variations of PC1 are dominated by the negative correlation between the lithogenic elements (Si, Ti and K), and S and Fe (Fig. 7). The positive correlation of Si with the detrital elements Ti (r=0.86, p<0.01) and K (r=0.84, p<0.01) indicates diatom productivity to play a minor role for changing silica

contents in KKJ sediments (Fig. 7). Although organic matter cannot be measured by XRF core scanning, the positive correlation between S and TOC (r=0.83, p<0.01) reveals that S is a proxy for organic matter accumulation in KKJ sediments (Fig. 8). Therefore, the S/Ti ratio can be used to indicate relative variations of organic and detrital material in KKJ19



sediments (Fig. 4). Moreover, our element clustering reveals a strong divide between the detrital and organic-rich sediments. The negative correlation between the elements Ca (and V) and Mn dominates the variations in PC2, which matches the
deviation found in the organic-rich sediments of clusters 3 and 4 (Fig. 7).

The four-cluster solution matches well with the main sedimentological changes described by the six SDU (Fig. 4). These four clusters can be related with the detrital sediments of SDU 1 (cluster 1) and the transition unit SDU 2 (cluster 2), as well as the organic-rich sediments of SDU 3 (cluster 3) and SDU 4 (cluster 4) (Fig. 4). The sediments in SDU 5 have a similar composition as in SDU 2 (both cluster 2), whereas the composition of the sediments in SDU 6 is similar to that in SDU 4
(both cluster 4). Organic sediments in cluster 3 are characterized by lower $\delta^{13}C_{org}$ values and TOC contents than those in cluster 4, as well as reduced variability (Fig. 8). Detrital sediments in cluster 1 contain less TOC than those in cluster 2, but about the same amount of S and Ti (Fig. 8). Sediments of SDU 3 (cluster 3) and SDU4 (cluster 4) are characterized by low amounts of Ti and higher amounts of S and TOC, while the amount of TOC is generally lower in sediments of cluster 3 than in those of cluster 4 (Fig. 8).


### 4.4. Chronology

A chronology for composite profile KKJ19 was constructed back to 9612 (+255/-144) cal. a BP (base of SDU 3 at 583 cm composite depth) using 8 [14]C dates from terrestrial plant macrofossils (Table 1), contents of the artificial radionuclides [137]Cs and [241]Am in the upper 31 cm and four marker horizons of known age (Fig. 9). Since no plant macrofossils were picked
below 536 cm composite depth (9009 +255/-144 cal. a BP), the [14]C based age-model was extrapolated to the base of SDU 3 considering the unit-wide stable sedimentation rates (13 years cm$^{-1}$ between the lowermost two [14]C samples) (Fig. 9). No age information is available for the detrital SDU 1 and 2.

Massive amounts of [137]Cs and [241]Am were released to the global atmosphere by surface nuclear weapon tests starting AD 1954 and peaking AD 1963 (Pennington et al., 1973). A second major release of [137]Cs mainly affecting the European
continent occurred during the Chernobyl Nuclear Power Plant accident in AD 1986 (Povinec et al., 2003). The associated fallout events are documented in KKJ sediments at 24.5 cm, 18.5 cm and 10.5 cm composite depth (Fig. 9). The [137]Cs and [241]Am-based age-depth model for the upper part of composite profile KKJ19 is independently validated by four marker horizons of known age: (I) The top of composite profile KKJ19 reflecting the coring year AD 2019, (II + III) two XRF Ca peaks at 3.5 and 5 cm composite depth indicating lake liming in AD 1999 and 1993, as well as (IV) an increase in detrital
matter input at 37 cm composite depth reflecting anthropogenic lake level lowering in AD 1878 (Figs. 2 and 9) (Stanton et al., 2010). In contrast to the results from the previous KKJ sediment core (Stanton et al., 2011), our microfacies analyses revealed no continuously varved sediments that could be used for chronological purposes for most of composite profile KKJ19.


## 5. Discussion

### 5.1. Holocene evolution of Lake Kälksjön

The succession of six SDU in composite profile KKJ19 can be interpreted in terms of different lake stages associated with changing environmental conditions and human impact in KKJ and its catchment.


*SDU 1 (pre-isolation phase):* Homogenous clay and fine-silt detrital material, as well as high Ti and low TOC contents suggest sediment deposition in an offshore location before the isolation of KKJ from ancient Lake Vänern (Figs. 4 and 5) (Björck, 1995). This is supported by the corresponding detrital element cluster 1, accompanied by the highest Ti and lowest TOC (~1%) contents of the record (Figs. 7 and 8). The material was most likely discharged into the basin during the
postglacial retreat of the Fennoscandian ice sheet (Risberg et al., 1996; Stanton et al., 2010).



*SDU 2 (transition phase):* A shift from a predominantly detrital to an organic deposition within SDU 2 is accompanied by the detrital element cluster 2, as well as decreasing Ti and increasing TOC contents, interpreted to reflect the transition towards a more local sediment source from the establishing catchment of KKJ (Figs. 4 and 7). Likely trigger of these

changes is the gradual isolation of the small lacustrine KKJ basin from ancient Lake Vänern through isostatic rebound and drainage of Lake Vänern via the Göta Älv into the Kattegat (Björck, 1995).

*SDU 3 (lacustrine sedimentation* 9612 (+255/-144) to 5434 (+78/-120) cal. a BP*):* The full isolation of KKJ at 9612 (+255/-144) cal. a BP at the base of SDU 3 is reflected by the onset of a continuous organic-rich sedimentation with benthic diatoms

and crysophyte cysts, as well as high TOC and low Ti contents (Figs. 4 and 5).

Our microscopically determined point of isolation is supported by that from Lake Skjutsbolstjärnet located at a similar location and height (98 m a.s.l., ~60 km southwest of KKJ), at 9600 a BP (Risberg et al., 1996). However, it is in contrast to that of the former KKJ sediment core at 9193 ± 186 a BP, defined by the onset of varve formation 30 cm above the onset of organic deposition at the base of SDU 3 (Stanton et al., 2010), that is not detectable in our microscopic observations.

However, considering the continuing drop in Ti at the base of SDU 3 until 9452 (+255/-144) cal. a BP to reflect this stratigraphic point would reconcile the 'isolation ages' from both KKJ sediment cores within errors (Fig. 4) (see Section 5.2 for a detailed discussion of SDU 3 and 4).

SDU 4 (lacustrine sedimentation 5434 (+78/-120) cal. a BP to AD 1878): Even though of similar composition, particularly

the TOC, Si/Ti and Ti records in SDU 4 reveal enhanced variability, compared to SDU 3 (Fig. 4). In addition, the change from SDU 3 to 4 is accompanied by a shift from the objectively determined element cluster 3 to 4 (Fig. 4) (see Section 5.2 for a detailed discussion of SDU 3 and 4).

*SDU 5 (lake level lowering,* AD 1878 to 1981 (+2/-3) a BP): A rapid increase in detrital sedimentation with high Ti and low

TOC contents in SDU 5 occurred concurrent with the onset of anthropogenic lake level lowering AD 1878 (Figs. 2 and 4). This rapid sedimentological change is paralleled by the return to the detrital element cluster 2, interpreted to reflect detrital sedimentation from a local sediment source (Fig. 4). The detrital material likely originates from the non-consolidated former shallow-water zone of KKJ. Few distinct detrital layers point to an event-based transport and deposition during snowmelt floods or heavy precipitation (Czymzik et al., 2010; Tiljander et al., 2003). The about 100-year-long duration of SDU 5

presumably indicates the time-span of soil formation and vegetation growth, under the influence of considerable land use. A sharp increase in settlement indicators (NAP upland and wild grass group pollen) and micro-charcoal particles at the onset of SDU 5 suggests that land use activity in the vicinity of the lake intensified concurrent with lake level lowering (Fig. 6).

*SDU 6 (post lake level lowering* AD 1981 (+2/-3) to 2019): Sediment composition and geochemistry in SDU 6 is similar to

that in SDU 4, suggesting a return to a relatively undisturbed lacustrine sedimentation (Fig. 4). Palynologic settlement indicators in KKJ sediments reveal a continuing human presence (Fig. 6).

**5.2. Mechanism of organic matter accumulation in SDU 3 and 4**

Within the organic-rich SDU 3 and 4, TOC contents in KKJ sediments reveal marked short and long-term variability (Fig. 4). Changing TOC contents in lake sediments are controlled by an interplay of productivity in the water column, supply from the catchment and post-depositional degradation (Meyers and Teranes, 2001). Comparing our TOC record during SDU



3 and 4 with the $\delta^{13}C_{org}$ and pollen data from composite profile KKJ19 allows us to disentangle the main control on organic matter accumulation in KKJ sediments.

$\delta^{13}C_{org}$ has been established as proxy for productivity in the photic zone of lacustrine systems (Meyers, 1994; Stuiver, 1975). The dissolved organic carbon pool of a lake becomes enriched in $^{13}C$, due to the preferential uptake of the lighter $^{12}C$ by phytoplankton (Teranes and Bernasconi, 2005). In consequence, phases of enhanced lake productivity are characterized by higher $\delta^{13}C_{org}$ values. The parallel increase in TOC and $\delta^{13}C_{org}$ in SDU 3 and 4, hence, suggests that the increasing organic matter accumulation is predominantly attributable to higher productivity in the water column (Fig. 4). A further major

influence on the $\delta^{13}C_{org}$ signature through varying inputs of allochthonous organic matter (Meyers, 1994) is unlikely, because of the microscopically determined rather homogenous composition of the organic sediment fraction during that period (Fig. 5). This is confirmed by C/N-ratios between 11 and 16 in SDU 3 and 4 indicating a rather stable and predominantly aquatic source of the organic material (Meyers, 1994) (Fig. 4).

To summarize, based on the covariance with the $\delta^{13}C_{org}$ record, we consider changes in the TOC record from KKJ to mainly

reflect varying in-lake productivity. We consider major human influences on productivity changes in KKJ during SDU 3 and 4 as unlikely since we only detect a minor increase in human settlement indicators (NAP upland and wild grass group pollen) at the transition of SDU 3 to 4, followed by a return to low values (Fig. 6). Also more distinct rises in these settlement indicators in KKJ sediments 500 cal. a BP and sediments of the nearby Lake Gloppsjön (~100 km west of KKJ) 2350 a BP (Almquist-Jacobson, 1994) are not paralleled by similar changes in our TOC and $\delta^{13}C_{org}$ records (Figs. 4 and 6).


### 5.3.1. Millennial productivity trend and orbital forcing
The multi-millennial upward trend in TOC contents (and simultaneously increasing $\delta^{13}C_{org}$ values) interpreted to reflect progressively increasing productivity in KKJ reveals a sharper increase until ~5450 a BP, followed by a more moderate rise until recent times (Fig. 10). This trend is paralleled by progressively warmer winter temperatures in north-western Europe

reconstructed from pollen records, interpreted to be mainly caused by a continuous increase in orbital Northern Hemisphere winter insolation (Davis et al., 2003; Laskar et al., 2004; Wanner et al., 2008) (Fig. 10). Enhanced productivity in KKJ associated with warmer winters is best explained by shortened ice cover (on average 5 months, today) allowing a prolonged growing season in spring and summer and increased metabolic rates (Karlsson et al., 2005; Willemse and Törnqvist, 1999). Winter temperature and coupled ice cover duration were reported as a main control of productivity from monitoring and

model studies of multiple Swedish lakes covering a wider latitudinal and altitudinal range (Blenckner et al., 2002, 2004; Karlsson et al., 2005) (see section 5.3.2. for details).

### 5.3.2. Decadal to centennial productivity variability and NAO
Today, meteorology at KKJ is correlated with the predominant mode of the NAO (Fig. 11). The NAO is the major source of

atmospheric circulation variability over the North Atlantic and Europe primarily during winter (Hurrell, 1995; Olsen et al., 2012). During its positive phase Scandinavian climate is characterized by above average temperatures, precipitation and windiness (Fig. 11) (Hurrell, 1995).

Multi-decadal to centennial productivity changes in KKJ during SDU 3 and 4 resemble changes in NAO polarity reconstructed from Greenland lake sediments back to 5200 a BP and tree rings for the Medieval Warm Period/Little Ice Age

transition (Fig. 12) (Olsen et al., 2012; Trouet et al., 2009). Productivity in the lake is higher, when the NAO is in a more positive polarity (Fig. 12). The common variability between TOC contents in KKJ sediments and NAO changes during the last 5200 years is supported by spectral similarities between both records, revealing broadly common oscillations around 150, 195, 230 and 270 years (Fig. 13) (Olsen et al., 2012). Minor spectral differences between both records might be connected to the uncertainties of the individual chronologies. Even though we rule out major human influences (see section





5.2), smaller deviations between the NAO reconstruction and TOC record from KKJ starting ~2350 a BP might be associated with the first occurrences of humans in the KKJ region since that time (Almquist-Jacobson, 1994).

In consequence, we interpret our decadal to centennial TOC record from KKJ sediments during the complete 9612 (+255/-144) years to mainly reflect qualitative changes in NAO-like atmospheric circulation. Analogue to the multi-millennial trend, main mechanistic linkage for the observed decadal to centennial TOC variability in KKJ might be the influences of the NAO

polarity on winter temperature, ice cover duration and lake productivity.

This interpretation is supported by meteorological and monitoring studies of several Swedish lakes indicating a significant influence of the NAO on annual to seasonal temperatures, ice cover duration and, consequently, productivity (Blenckner et al., 2004; Chen and Hellström, 1999; Karlsson et al., 2005). Winter temperatures in Sweden are warmer, ice cover is shortened and productivity higher, when the NAO is in a positive mode (Blenckner et al., 2004; Chen and Hellström, 1999;

Hurrell, 1995). The importance of the NAO for ice cover duration can be exemplarily described by monitoring results from three lakes in vicinity to KKJ (all 60°N in Sweden) covering the period AD 1961 to 2002 (Blenckner et al., 2004). Mean ice cover duration of the three lakes within this time-interval varies between 99 and 203 days and the NAO is one significant driver of the up to >3 months changes in ice cover duration (Blenckner et al., 2004; Ptak et al., 2019). In addition, NAO influences on ice cover duration of Swedish lakes are particularly strong south of 62°N where KKJ is located since the

blocking of the North Atlantic zonal atmospheric circulation by the Scandinavian Mountains is minor (Blenckner et al., 2004).

Further influences of changes in Siberian High (SH) strength on productivity in KKJ are possible, since the lake is situated at the western boundary of this atmospheric system. However, on the one hand, meteorological investigations and paleoclimate reconstructions indicate that NAO and Siberian High changes are interdependent, particularly on long time-scales (Fig. 12).

Siberian High strength tends to be reduced when the NAO is in a more positive mode (Chen et al., 2010; He et al., 2017). On the other hand, KKJ is located in direct vicinity to the North Atlantic within the path of the westerly storm tracks. Therefore, considering the location of KKJ and intercontinental teleconnections, we prefer to relate the decadal to centennial changes in TOC as driven by NAO-like changes in atmospheric circulation.

Interestingly, decadal to centennial productivity changes in the KKJ sediment record depict a regime shift towards larger

variability starting concurrent with the onset of Neoglaciation ~5450 cal. a BP (Fig. 12). This shift in TOC is accompanied by the change from SDU 3 to 4 and from element cluster 3 to 4 (Fig. 4). It coincides with an increase in valley floor incision in the southern German Lech catchment (Köhler et al., 2022) and the onset of flood layer deposition in the sediment record from pre-alpine Lake Ammersee (Czymzik et al., 2013). It is broadly synchronous with a depletion of deuterium isotopes in Lake Torneträsk sediments (North Sweden) and changing pollen assemblages in sediments of two lakes from central Sweden

(Giesecke, 2005; Thienemann et al., 2018) (Fig. 12). All changes in the latter five central-northern European proxy records were interpreted to reflect a regime shift in atmospheric circulation. The two pollen records further rule out an anthropogenic origin of the signal (Giesecke, 2005). In addition, the recorded change in TOC variability coincides with the onset of strengthened variability in an isotope record from Lake Bjärsträsk located east of KKJ (Gotland Island), interpreted to reflect Siberian High strength (Fig. 12) (Muschitiello et al., 2013).

Based on the interpretation of TOC variability in KKJ sediments as predominantly driven by NAO polarity changes on decadal to centennial scales, we hypothesize the described European climate shift at ~5450 cal. a BP as also associated with a reinforcement of NAO-like atmospheric circulation variability. Proposed trigger of strengthened NAO polarity changes is a more variable North Atlantic deep-water formation (Olsen et al., 2012; Trouet et al., 2012). This is supported by a record of Holocene North Atlantic deep water formation deduced from $\delta^{13}C$ values of benthic foraminifera in a marine sediment core,

revealing enhanced variability since ~5450 a BP (Repschläger et al., 2015).



## 6. Conclusions

Holocene sediments from KKJ provided insights into the stages and timing of lake evolution associated with postglacial

landscape evolution, human interferences and climate variability in west-central Sweden. Following the isolation from ancient Lake Vänern through isostatic rebound 9612 (+255/-144) cal. a BP, varying TOC contents in KKJ sediments are interpreted to predominantly reflect changes of in-lake productivity modulated by the influences of winter temperature variability on ice cover duration and growth season length. An exception is the period from AD 1878 until today, in which sedimentation in KKJ was dominated by anthropogenic lake level lowering and land use. Productivity increases in KKJ

sediments are likely driven by the progressive millennial-scale winter warming in north-western Europe, following the increasing Northern Hemisphere winter insolation, and decadal to centennial periods of a more positive NAO polarity. Strengthened productivity variability since ~5450 cal. a BP is hypothesized to reflect a reinforcement of NAO-like atmospheric circulation, concurrent with the onset of more vigorous variations in North Atlantic deep-water formation. Our results reveal the importance of terrestrial Holocene paleoclimate records for disentangling the time-transgressive and non-

linear interplay of internal and external forcing on regional climate variability.


**Data availability**

All original data from this study will be published in the PANGAEA data library.

**Author contribution**

MC initiated the study, carried out microfacies analysis and led the writing. RT carried out XRF measurements and statistical

analyses. BP and SP performed geochemical measurements. PF conducted the hydroacoustic survey. MT performed pollen analyses. MM measured [137]Cs and [241]Am contents. MC, RT, BP, SP, PF, MT, MM, MJS, CKMN, AB and HWA contributed to the interpretation of the data and writing of the final paper.

**Competing interests**

The authors declare that they have no competing interests.

**Acknowledgements**

We thank Brian Brademann (GFZ) and Hendrick Mück (IOW) for their help during the coring campaign at Lake Kälksjön. Michael Köhler (MKfactory) produced thin-sections. MC is financed through grant CZ 227/4-1 (SyncBalt project) of the

German Science Foundation (DFG). This publication is a contribution to the BaltRap project SAW-2017-IOW-2, funded by the Leibniz Association.




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

**Table 1. Radiocarbon ($^{14}$C) ages of terrestrial plant macrofossils from composite profile KKJ19 with sediment depths.**

| Laboratory number | KKJ19 composite depth (cm) | Dated material | AMS $^{14}$C age (a BP ± 2σ) |
|---|---|---|---|
| Beta - 592289 | 23 | plant | 240 ± 30 |
| Beta - 584480 | 107 | plant | 1870 ± 20 |
| Beta - 568243 | 151 | wood | 2530 ± 30 |
| Beta - 568244 | 176 | wood | 3080 ± 30 |
| Beta - 592290 | 239 | plant | 4180 ± 30 |
| Beta - 584482 | 314 | plant | 5130 ± 30 |
| Beta - 592291 | 429 | plant | 6820 ± 30 |
| Beta - 584481 | 536 | plant | 8050 ± 40 |



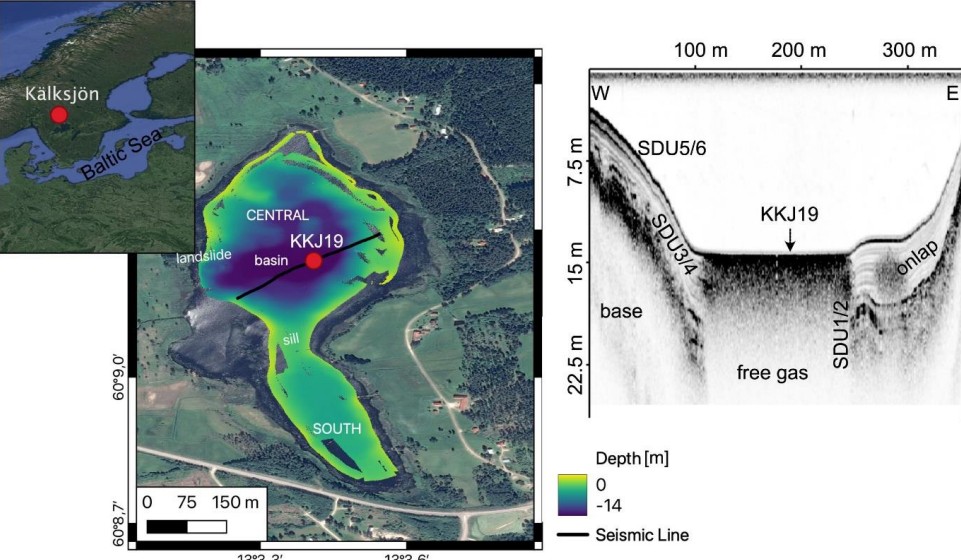

**Figure 1: Geographical location of Lake Kälksjön (KKJ) in the western Baltic region and bathymetric map of the basin (©2015 Google Maps). In- and outflows, as well as the coring location of composite profile KKJ19 are indicated. A seismic line crossing the coring location displays the sedimentary sequence of KKJ. The expected seismic reflections of the six sediment deposition units (SDU) in KKJ19 are indicated.**

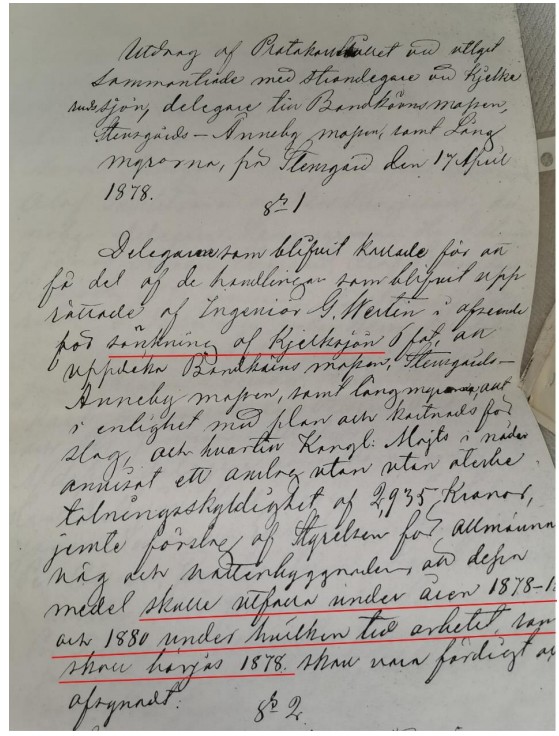


**Figure 2: Historic document indicating that lake level lowering of Lake Kälksjön (avsänkning of Kjelksjön) should be performed within the years AD 1878 until 1880 (skulle utföra under aren 1878-1??? och 1880…) and that the work should start in AD 1878 (…under vilken tid arbetet som skulle börjas 1878).**




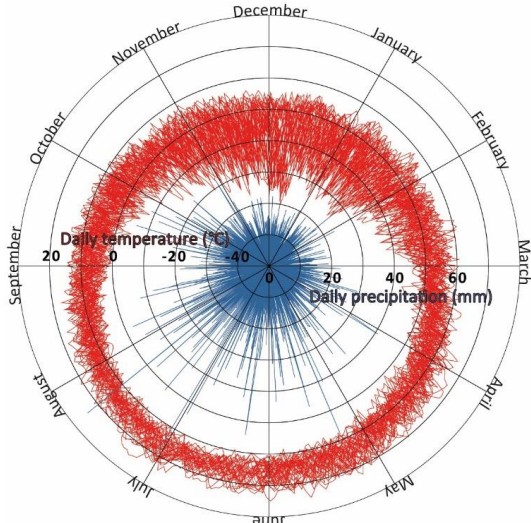


**Figure 3: Daily temperatures from AD 1961-2012 and daily precipitation from AD 1945-2013 recorded at the SMHI station Torsby, located ~5 km west of Lake Kälksjön (KKJ).**


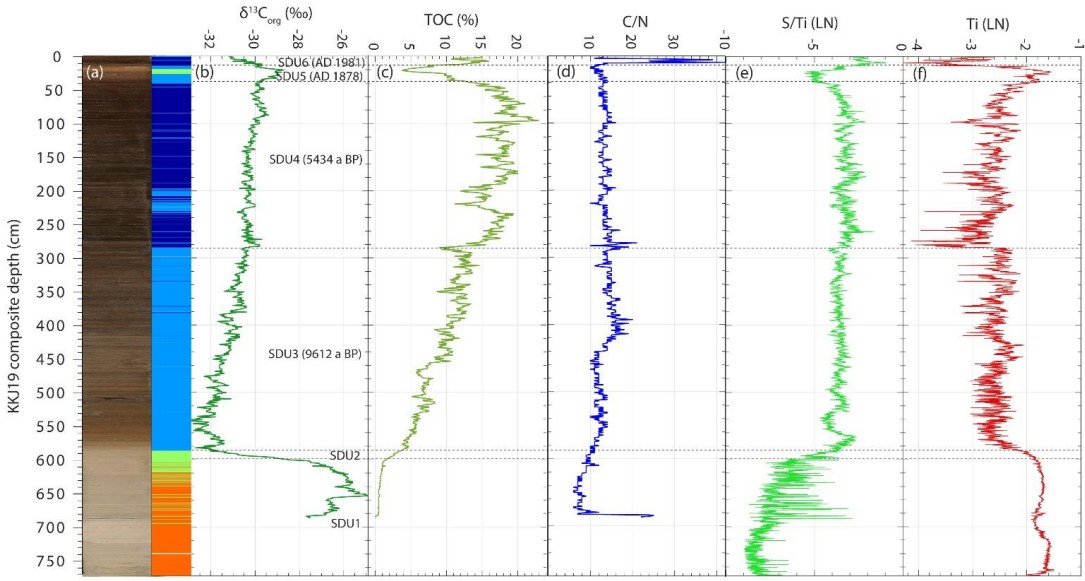

**Figure 4: Composite profile KKJ19 with selected proxy records. (a) Photograph of composite profile KKJ19 with**
**XRF-based element cluster stratigraphy and corresponding (b) $\delta^{13}C_{org}$, (c) total organic carbon (TOC), (d) C/N, (e) S/Ti and (f) Ti records. Sediment deposition units (SDU) 1 to 6 are indicated. Dates for the onsets of SDU are given, when applicable.**



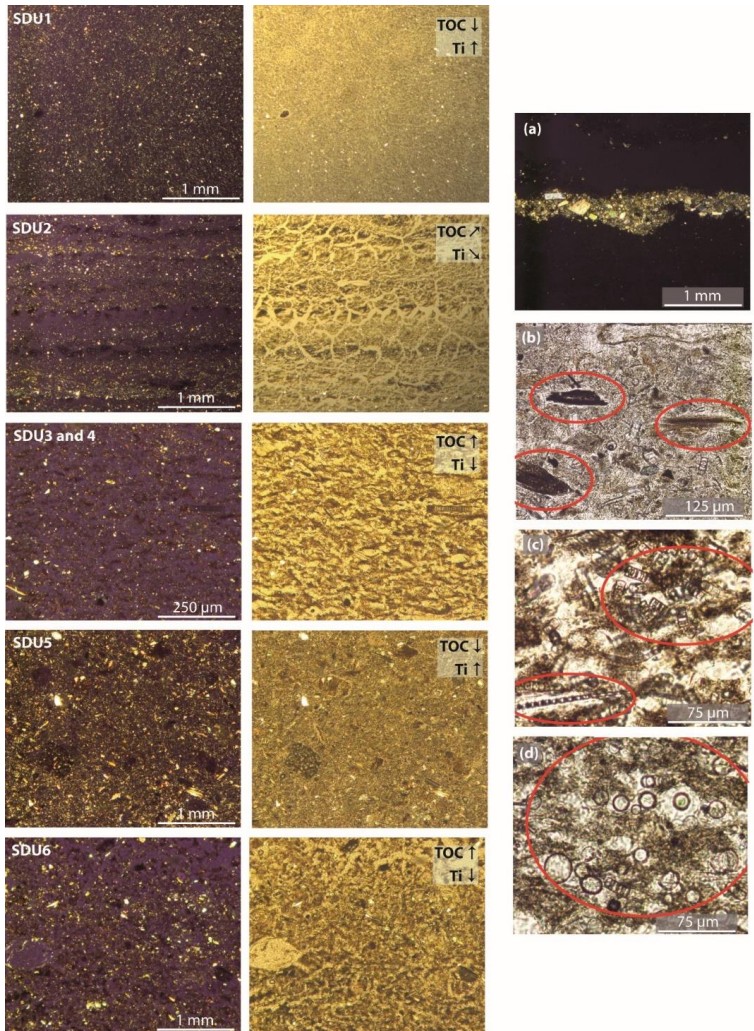

**Figure 5: (left) Overview micrographs of Lake Kälksjön (KKJ) sediment deposition units (SDU) 1 to 6 under polarized and plain light. Relative abundances of TOC and Ti contents in each SDU are indicated. (right) Micrographs highlighting individual sediment components. (a) Liming layer AD 1993 (5 cm composite depth, polarized light), (b) terrestrial organic debris in SDU 5 (18 cm composite depth, plain light), (c) diatom frustules (Aulacoseiraceae and Fragilariaceae) in SDU 3 (305 cm composite depth, plain light), (d) crysophyte cysts in SDU 3 (298 cm composite depth, plain light).**



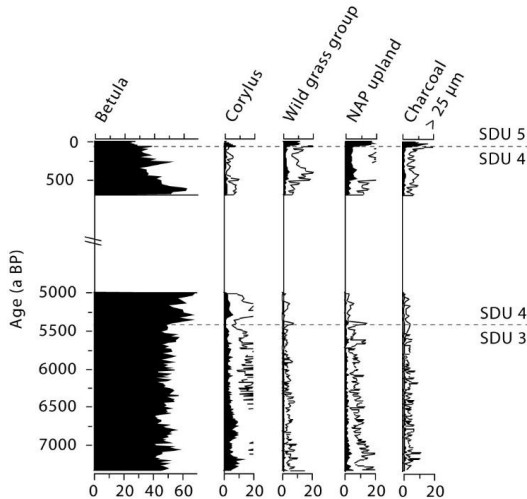

**Figure 6: Pollen records from Lake Kälksjön (KKJ) sediments covering the transitions between SDU 3 and 4, as well as SDU 4 and 5.**

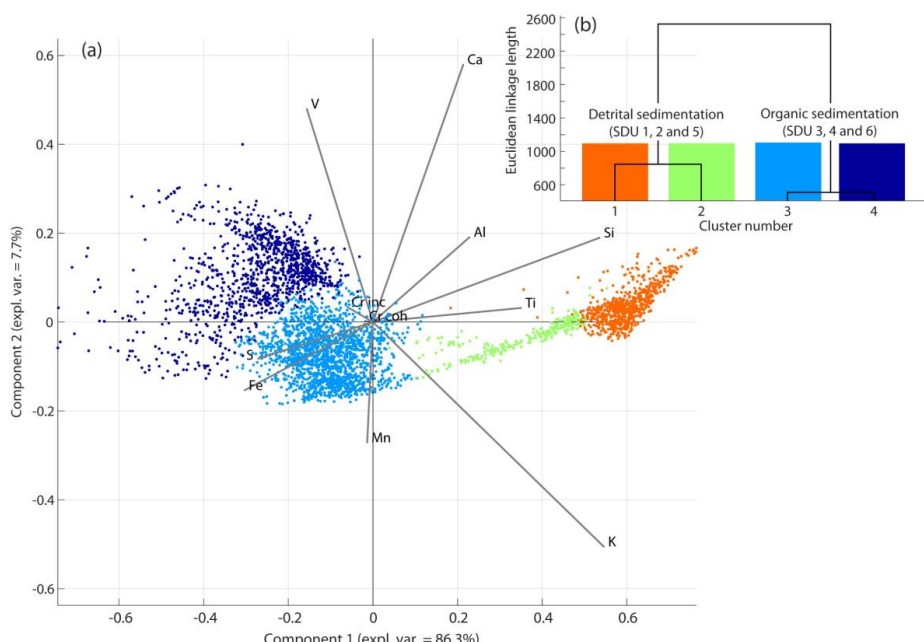

**Figure 7: Hierarchical clustering of XRF profiles from Lake Kälksjön (KKJ) sediments. (a) Covariance biplot visualizing the correlations of the main elements with regard to the first two principle components. (b) Element clusters reflecting significantly different sediment compositions.**





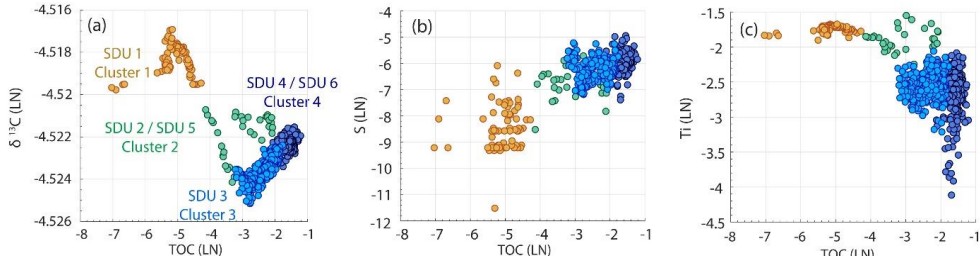

**Figure 8: Comparison of bulk geochemistry and XRF data from composite profile KKJ19. (a) δ¹³C of organic**
**material (δ¹³C_org) and total organic carbon (TOC). (b) S and TOC. (c) Ti and TOC. Colours of the data points**
**correspond to those of the cluster analysis.**

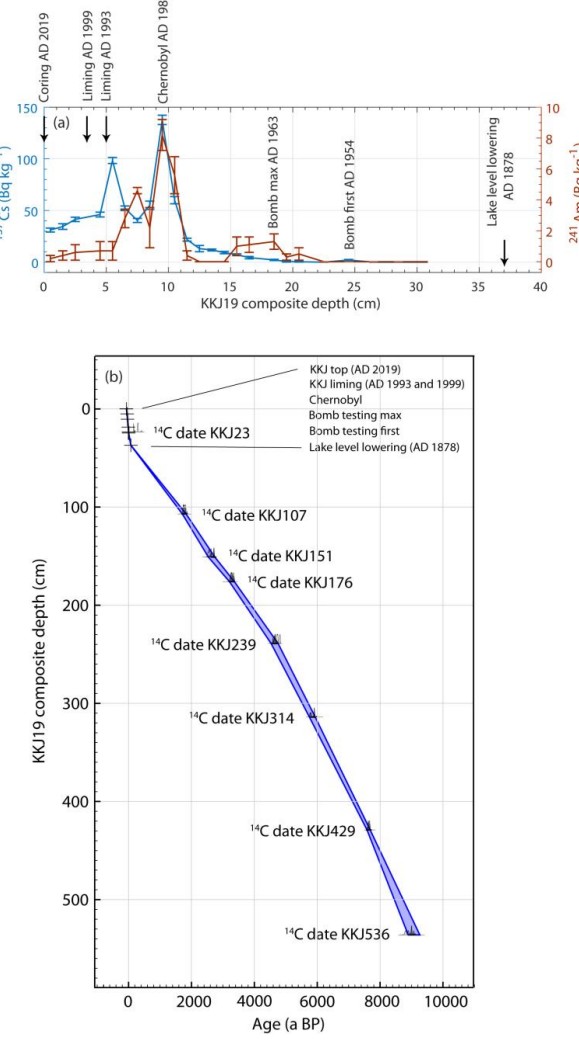

**Figure 9: Age-depth model for composite sediment profile KKJ19. (a) ¹³⁷Cs and ²⁴¹Am dating of the topmost**
**sediments and positions of lithological marker layers of known age. (b) Calibrated ¹⁴C dates from 8 plant**
**macrofossils. The age-depth model was constructed using the OxCal software working with the IntCal20 calibration**
**curve (Bronk Ramsey, 2008; Reimer et al., 2020).**





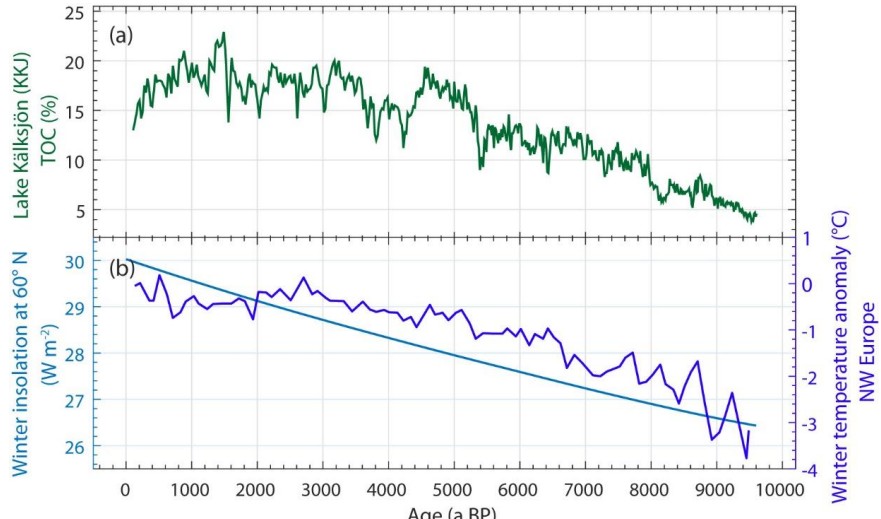


**Figure 10: Multi-millennial trends in paleoclimate and orbital forcing during the last ~9600 years. (a) Lake Kälksjön (KKJ) total organic carbon (TOC) contents. (b) Winter (January) insolation at 60°N (Laskar et al., 2004) and winter temperature anomaly in north-western Europe (Davis et al., 2003).**


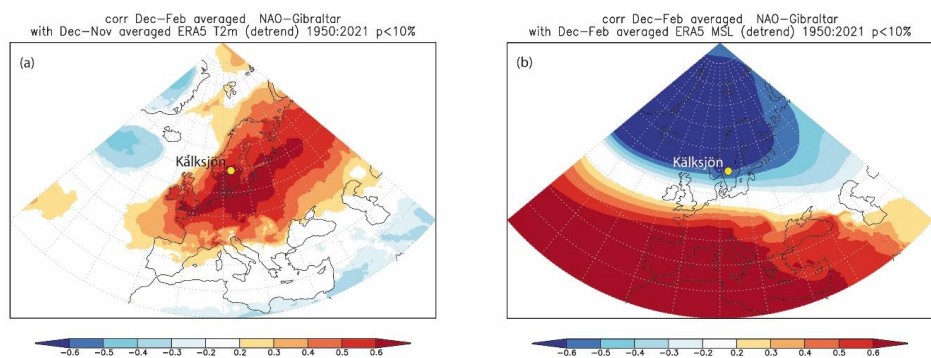

**Figure 11: Correlation of North Atlantic Oscillation index (Jones et al., 1997) with (a) sea level temperature and (b)**
**sea level pressure from the ERA 5 reanalysis (Hersbach et al., 2020) for the period 1950 to 2021 during winter (DJF).**



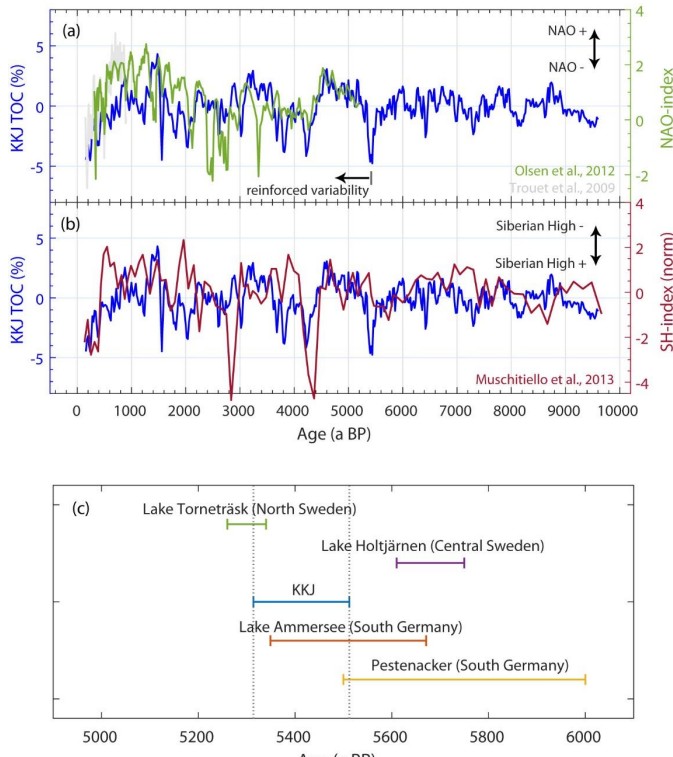

**Figure 12: Decadal to centennial paleoclimate records covering the last ~9600 years. (a) Total organic carbon (TOC)**
**record from Lake Kälksjön (KKJ) along with reconstructions of the North Atlantic Oscillation (NAO) (Olsen et al.,**
**2012; Trouet et al., 2009). (b) TOC record from KKJ along with proxy of Siberian High strength (Muschitiello et al.,**
**2013). Multi-millennial variability was removed from the KKJ TOC record by subtracting a 2500-year low-pass**
**filtered version from the original time-series. (c) Timing of regime shifts in atmospheric circulation from north-**
**central European paleoclimate records during the proposed reinforcement of NAO variability imprinted in KKJ**
**sediments (Czymzik et al., 2013; Giesecke, 2005; Köhler et al., 2022; Thienemann et al., 2018).**





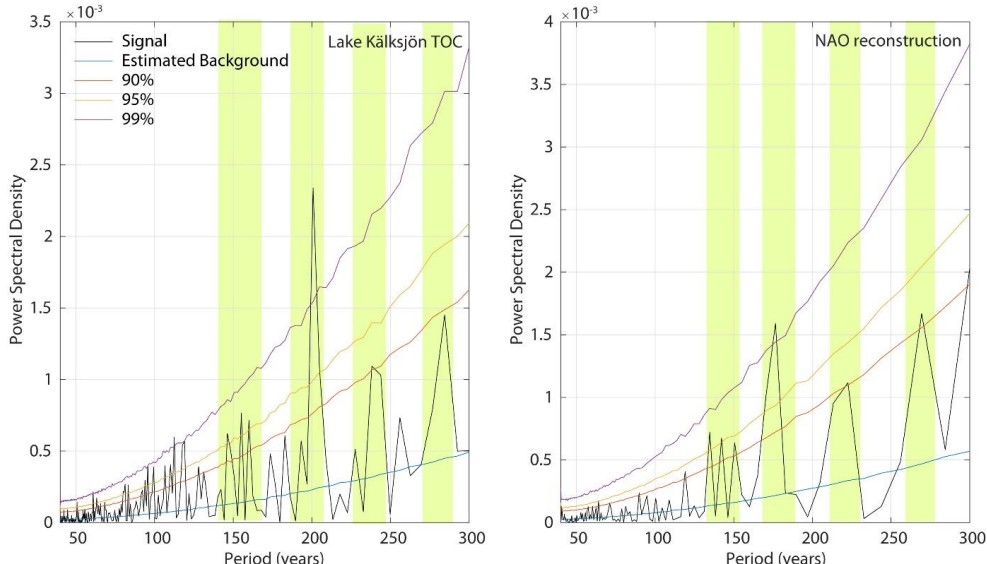

**Figure 13: Spectral analyses of total organic carbon (TOC) record from Lake Kälkjsön (KKJ) sediments and a multi-millennial NAO reconstruction (Olsen et al., 2012). Before the analyses, both records were resampled to a 20-year**
**resolution and detrended. Significance levels were estimated using 10000 iterations of an autoregressive model fitted to the original time-series.**
