# Peer review of "Mid-Holocene reinforcement of North Atlantic atmospheric circulation variability from a western Baltic lake sediment record"

_EGUsphere, 2022_

## Author Comment (AC2)

**Response to the reviewers' comments**

We thank the two reviewers and editor Denis-Didier Rousseau for their constructive and detailed comments to our manuscript. In the following we will respond to all concerns raised, first answering the main point of criticism, and then in a point-by-point reply.

**Detailed response (qualitative NAO polarity changes in TOC from KKJ sediments):**

We agree with Referee #1 about the non-stationary behavior of changes in the North Atlantic Oscillation (NAO) and the problems of covering these changes in individual records. To respond to the Referee's comments and perform the requested further examinations, we now compare our TOC record from KKJ sediments with 5 different NAO reconstructions, updated the connected Fig. 12 and added new paragraphs to chapter 5.3.2., discussing the broad covariance and remaining disagreements between the KKJ TOC and NAO records, as well as the possible underlying mechanisms (see the revised chapter and Fig. 12 below). However, despite the resulting visible matches, the correlation coefficients between the KKJ TOC record and NAO reconstructions >1000 years (all sampled to 20-year resolution) are low between 0.14 and 0.32 with significance levels between 0.05 and 0.2. Most likely and important reason are the uncertainties of in particular the individual radiocarbon-based chronologies, allowing to allocate shorter-term variability and hampering the statistical 1 to 1 matching of these changes. For example, the average chronological uncertainty of the investigated KKJ sediment record is ±79 years. Considering this technical limitation, we think that correlation coefficients and significance levels do not reflect a realistic degree of similarity between the records and prefer not to show them.

The revised chapter (red text is new):

**5.3.2. Decadal to centennial productivity variability and NAO**

[revised manuscript text omitted]

**Revised Fig 12. The revised figure now includes a comparison of the KKJ TOC record with 5 different NAO reconstructions and an index of Siberian High strength (a-f). In (g) we added the onset of reinforced paleoclimate variability at ~5450 a BP recorded in the nearby Store Mosse bog. We will**

**add the archive-type (e.g. tree rings) and geographical position of the compared NAO records to**
**the figure caption.**
**Point by point response to Referee #1**

This is a very interesting study which attempts to provide insights into
the North Atlantic Oscillation (NAO) variability throughout most of the
Holocene. The detailed sedimentology from lake Kalksjon, west-central
Sweden, is excellent and provides some justification for its use as a
qualitative reconstruction of the NAO.  Overall, I found the manuscript
well written and will be of high interest for the paleoclimate community
studying the climate during the Holocene for this region. It is therefore
suitable for Climate of the Past. However, to assess this archive as a
reconstructed "qualitative" record of the NAO, I believe there needs to be
further examinations.

Thank you!

Major comments:

Different NAO reconstructions show periods of coherency and often no
coherency at all. This is partly because the NAO itself exhibits non-
stationary behavior, so the use of one single location may not capture the
whole variability. One aspect that may have been overlooked in this paper
is the other existing NAO proxies. Have you explored other records that may
be sensitive to large-scale and long-term NAO fluctuations? That said, from
1800BP to the present, the NAO from Olsen et al. (2012) and the TOC record
don't seem to match quite well, but perhaps if you plot different
reconstructed NAO, a better co-variability (correlation) may be seen. I am
thinking of speleothems in Europe and North Africa as well as the new one
from Becker et al. (2020). See also Wassenburg et al. 2016 Nat Geosci,
Baker et al. 2015, etc.

Please see our Detailed Response.

Figure 12 shows the relationship between paleo NAO (Olsen et al. 2012) and
the TOC content in the studied lake. When sampling both records to the
lowest resolution of the corresponding record; do you find any significant
correlations? Same comment for the Siberian High.

Please see our Detailed Response.

Also, the lack of coherence between your record and other NAO proxies could
be explained by other mode of variability that may have been more
persistent in the past. The Scandinavian Blocking, for example, accounts
for ~27% of the winter North Atlantic variability. A persistent
Scandinavian Blocking in winter would translate to cooler conditions in the
region, thereby presumably increased ice cover time (in turn less productivity). Any thoughts on this?  I suggest to add more discussion
around line 345.

Please see our Detailed Response.

Some other minor comments:

Figure 3: Given you are dealing with decadal to centennial scale
variability, I think an average (monthly?) of temperature and precipitation
would improve visualization.

We revised Fig. 3 following the referee's suggestions:

[Figure]

**Revised Fig. 3 now showing monthly averaged precipitation and temperature data from the**
**meteorological station Torsby (~5 km west of Lake Kälksjön) for the observation period.**

Figure 4 shows the composite with the XRF cluster stratigraphy. It should
be located after figure 7 (PCA of the elements).

We agree with Referee #1 that the XRF cluster stratigraphy in Fig. 4 should be in theory
located after Fig. 7. However, without the direct comparison with the $\delta^{13}$C, TOC, C/N, S/Ti
and Ti proxies from KKJ sediments a lot of information about the cluster stratigraphy and the
connected sedimentology would get lost. Therefore, we would like to keep the cluster
stratigraphy in Fig. 4. To account for the reviewer's comment, we added to the text '(see Fig.
4)', when the XRF cluster stratigraphy is mentioned.

Additionally, we slightly modified the caption of Fig. 7 into:

**Figure 7: Statistical analyses of the XRF profiles from Lake Kälksjön (KKJ) sediments. (a) Covariance**
**biplot visualizing the correlations of the main elements with regard to the first two principal**
**components. (b) Hierarchical clustering solution reflecting the difference between the detrital**
**sediments of SDU 1, 2, and 5 and organic sediments of SDU 3, 4, and 6.**

Figure 4: Add average temporal resolution for each cluster, i.e., mm/year

To improve/keep the readability of Fig. 4, we added the requested information to section
5.1. (Holocene evolution of Lake Kälksjön). Only the names of the SDU remained as text in
the figure. Please see also the related comment to Referee #2, requesting SDU age ranges in
Fig. 4.

Figure 10: the sharp decline in TOC falls within 4.2k BP. Do you consider
that your proxy responded to the 4.2k event?

Considering the complete TOC record from KKJ sediments shown in Fig. 12, the drop in TOC
at 4.2 ka is not unique and appears somewhat short for marking the coinciding climate
event. Therefore, we would like to avoid a discussion about the connection of this peak to
the 4.2 ka climate event.

Figure 12: we don't see much the Trouet et al. NAO, the color is too pale.
Also, why not the selection of the Ortega et al. reconstructed NAO?

Please see our Detailed Response and the revised Fig. 12. We omitted the NAO
reconstruction by Ortega et al. (2015) for the last millennium since it does not reflect the
shift from a more negative to more positive NAO polarity between the Little Ice Age and
Medieval Warm Period that is the most distinctive last millennium feature in the other 5
NAO reconstructions used in this study.

Figure 12: the lines of the SH index are outside the x-axis.

We corrected that.

Figure 13: the spectral peaks don't seem to match quite well, perhaps a
cross-spectral analysis would give something better. I would suggest moving
this to the supplement.

Considering the non-stationarity of the NAO suggested by the Referee, we prefer not to
show spectral analyses or cross wavelet analyses in the updated manuscript.

Line 89: could you provide more information as to how the grain-size was
extracted?

The particles-sizes of individual detrital grains were measured with a scale under the
microscope. We added 'microscopic' to the text.

Section 3.3: provide information on how many thin-sections were produced.

We added to the text that 80 thin-sections were produced from KKJ sediments.

Section 3.5: Why no pollen analysis on other SDU?

Pollen analysis were performed on KKJ sediments to cover the transition between SDU 3 and
4 (transition to higher amplitudes in the TOC record) and the recent period of most distinct
human activity. For the remaining periods without exceptional variability in KKJ TOC record,
we believe that it is appropriate to refer to pollen results from nearby lakes to rule out major
human influences.

Section 4.3: Add a Table showing the matrix correlation between μ-XRF data

We added the following table with a correlation matrix for the XRF data from KKJ sediments
to the manuscript:

**New Table. Correlation matrix for the XRF profiles from the Lake Kälksjön KKJ19 composite profile.**

|        | Al  | Si   | S     | K     | Ca    | Ti    | Mn    | Fe    |
|--------|-----|------|-------|-------|-------|-------|-------|-------|
| **Al** | 1   | 0.95 | -0.92 | 0.81  | 0.75  | 0.87  | -0.37 | -0.83 |
| **Si** |     | 1    | -0.92 | 0.90  | 0.71  | 0.90  | -0.28 | -0.92 |
| **S**  |     |      | 1     | -0.88 | -0.70 | -0.91 | 0.14  | 0.77  |
| **K**  |     |      |       | 1     | 0.45  | 0.91  | -0.04 | -0.83 |
| **Ca** |     |      |       |       | 1     | 0.61  | -0.46 | -0.68 |
| **Ti** |     |      |       |       |       | 1     | -0.35 | -0.82 |
| **Mn** |     |      |       |       |       |       | 1     | 0.25  |
| **Fe** |     |      |       |       |       |       |       | 1     |

Line 104: 3s is deem low. Is this a typo?

3 seconds is correct. Using the latest XRF detector in combination with high electric current
(60 mA) allows measuring with such low acquisition time.

Line 105: why only these elements? For example, why is Ca omitted?

The elements mentioned in line 105 are selected based on the amount of zero-values and
replicate measurements. This is now explained in lines 102-107.

Line 106: How did you build the µ-XRF composite? Elements often decrease
(increase) their values at both edge of the sediment sections. Did you
remove those data?

The XRF data were measured for composite profile KKJ19 omitting sediment core endings.
We added to the text that the XRF data were acquired for composite profile KKJ19.

Line 239: There is no Ca profile showing these peaks. Maybe add into
supplement.

Instead of adding an additional figure with Ca, we now refer to the microscopically visible
liming layers at 3.5 and 5 cm composite depth. A micrograph of one of these layers is shown
in Fig. 4.

Line 364-365: Please rephrase

The rephrased sentence:

Decadal to centennial productivity changes revealed by the KKJ sediment record indicate a
shift towards reinforced variability concurrent with the onset of Neoglaciation ~5450 cal. a
BP (Fig. 12).

Thank you very much!

**Point by point response to Referee #2**

I agree with reviewer 1 that this is a very interesting and well-written
paper that provides new insight into NOA variability using a Swedish lake
record. I am, however, not an expert in NAO and NAO variability so I have
focused my review on other aspects of the paper.

Thank you!

Comments:

Line 18    Maybe a minor comment, but I think that LKJ would be a better
acronym than KKJ

We prefer to continue using KKJ as acronym for Lake Kälksjön. The works on this sediment
archive in different laboratories since the coring in 2019 were performed using KKJ. A
changed acronym might cause confusion during future work on this sediment archive.

Line 19    I prefer CE (Common Era) rather than AD

We replaced AD with CE in the text and figures.

Line 42    Here it might be useful if you define what you mean with the
western Baltic region (or the western Baltic Sea region?). There are at
least ten different definition of the "Baltic Region" in Wikipedia

We now define in line 35 that our 'Western Baltic region' comprises 'south-central
Scandinavia'.

Line 55    I checked the Swedish Land Survey online maps and long. 13°03'E
is more correct.

Thank you. We changed this information accordingly.

Line 59    west

Corrected.

Line 63    Figure 3 shows daily temperatures and precipitations, not mean
monthly temperatures, see also comment by Rev. 1

The revised Fig. 3 now shows mean monthly temperature and precipitation values for the
observation period at the SMHI station Torsby. Please see the revised figure in our response
to Referee #1.

Line 105   I agree with Rev. 1, why was not e.g. Zr and Ca analysed.

As stated before, the elements mentioned in line 105 are selected based on the amount of
zero-values and replicate measurements. This is now explained in lines 102-107.

Line 228   I would use "concentrations of the artificial radionuclides"
rather than "contents"

We replaced 'contents' with 'concentrations'.

Line 238 See comment by Rev. 1. Was Ca measured after all?

Please see the comment above. Yes, Ca was measured. A positive correlation with the
elements Ti and K indicates its predominantly detrital origin.

Line 266  I was slightly confused by the discussion about the isolation of
the lake basin from Ancient Lake Vänern. I have not read the Risberg et al
paper in any detail, but the uplift history of the area is complicated with
evidence of irregular postglacial isostatic uplift. Hence, the comment that
the isolation of L. Skjutsbolstjärnet located at a similar location (and
what is meant by that?) and height supports your isolation point must be
clarified (or deleted?). The isolation age of L. Skjutsbolstjärnet is c.
9600 uncalibrated C14 years which calibrates to c. 10,900 cal a BP

Based on the comment of Referee #2 on the irregular uplift history of the region around KKJ,
we deleted the information about the isolation age of Lake Skjultsbolstjärnet.

Line 269 cal is missing before "a BP" (Stanton et al. 2010)

We added cal.

Line 313  typo? Should be east of L. Kälksjön

We corrected the typo.

Line 318  My alternative interpretation of the TOC contents would be "a
sharp increase until ca 4500 cal a BP with a drop at ca. 5300 cal a BP",
i.e. the increase continues another 1000 years, followed by more stable
values until recent times. I would also add $\delta^{13}$C to Figure 10.

As suggested by Referee #2, the revised Fig. 10 now also shows $^{13}C_{org}$ along with TOC from
KKJ sediments.  Comparing both records from KKJ sediments indicates the difficulties of
identifying a certain change-point in its multi-millennial behavior. Therefore, we slightly
modified the connected sentence to:

The multi-millennial upward trend in TOC contents (and simultaneously increasing $\delta^{13}C_{org}$
values) interpreted to reflect progressively increasing productivity in KKJ reveals a sharper
increase until the Mid-Holocene, followed by a more moderate rise until recent times.

[Figure]

**Revised Fig. 10 now also showing $^{13}C_{org}$ from KKJ sediments.**

Line 331  Does this refer to winter or annual temperatures, precipitation
and windiness?

We added 'winter' to the related sentence.

Line 341  I am not certain that the reference to Almquist-Jacobson is 100%
correct here. Almquist-Jacobson's sites are situated in an area that may
have been settled by humans later than the river and lake valleys in
Värmland. There are archaeological findings north and north-west of Torsby
dating back to the early Neolithic (3800-3300 BC)

The thank the referee about the information on early Neolithic findings north and north-
west of Torsby. However, we did not find coincidences between changes in the TOC record
and changes in pollen indicators of human activity directly from KKJ sediments during this
early Neolithic period.

We replaced the reference to Almquist-Jacobsen with the one by Eddudottir et al. (2021)
finding an increase in human activity at about the same time (2100 a BP) in the much more
nearby Lake Karebolssjön (~25 km northeast of KKJ). Still, there is no corresponding change
in the TOC record from KKJ sediments.

Line 364-374    Here you should also discuss proxy records from Sweden
showing climate shifts at this time, e.g. the peat record from Store Mosse
(Kylander et al., 2013; QSR) and lake level records from L. Bysjön
(Digerfeldt, 1988; Boreas). These could also be shown to Fig. 12.

We agree with Referee #2 about the importance of adding results from the regional
paleoclimate records of the Store Mosse bog and Lake Bysjön to our discussion on reinforced
NAO-like atmospheric variability since ~5450 a BP.  Supporting our interpretation, the PC4
dust time-series from the nearby Store Mosse bog reflecting wind changes depicts enhanced
variability since this time. We added this information to the revised Fig. 12 and the text.
Enhanced shorter-term variability since about 5450 a BP is also present in the lake level
curve from Bysjön. However, the low resolution of the record inhibits detecting a certain
change-point and possible high-amplitude variations. Therefore, we mention the result of
shorter-term lake level variability since about 5450 a BP only in the text.

Line 546  Change Väners to Vänern

Done.

Table 1    If possible, give weights and type of material (e.g. terrestrial
or lacustrine plant remains?)

We added to Table 1 that all [14]C dated material is terrestrial. The weights of the samples are
unknown. But, BETA rejects samples before the [14]C measurement, if they are too small.

Figure 2  Give the source of the historic document. Museum, library?

We added to the caption of Figure 2 that the document was provided by a local historian.

Figure 4   Sediment deposition units. Give the age range for each SDU, if
possible, not only the onset

Please see also our comment on Fig. 4 above. To improve/keep the readability of Fig. 4, we
added all requested information to section 5.1. (Holocene evolution of Lake Kälksjön). Only
the names of the SDU remained as text in the figure.

Thank you very much!